# An Experimental Evaluation of Respiration by Monitoring Ribcage Motion

Marco Ceccarelli [1,*] , Manuel D'Onofrio [2], Vincenzo Ambrogi [2] and Matteo Russo [1]

1   Department of Industrial Engineering, University of Rome Tor Vergata, Via del Politecnico 1,
   00133 Rome, Italy; matteo.russo@uniroma2.it
2   Department of Surgical Sciences, University of Rome Tor Vergata, Via Montpellier 1, 00133 Rome, Italy;
   manuel.donofrio@students.uniroma2.eu (M.D.); ambrogi@uniroma2.it (V.A.)
*   Correspondence: marco.ceccarelli@uniroma2.it

**Abstract:** This paper aims to make an early diagnosis of respiratory disorders by measuring ribcage motion. A statistically significant numerical evaluation of the biomechanics of respiration is obtained through the acquisition of the kinematics of the sixth rib. We report the results of an experimental campaign that has been conducted using a RESPIRholter prototype for efficient and comfortable respiration monitoring on two groups of volunteers, one with healthy people and the other with chest-operated patients. The data from repeated acquisitions are statistically processed to analyze results in terms of angular motion and linear acceleration, which can be used to characterize and classify respiration motion. This experimental campaign can be considered a first result for the construction of a database useful for a reference of diagnostics, as reported by the discussed example case study.

**Keywords:** biomechanics; respiration; experimental evaluation; procedure; statistical elaboration

## 1. Introduction

This paper presents the results of an experimental investigation for analyzing ribcage motion during respiration with a portable RESPIRholter device that is based on an IMU (Inertial Measuring Unit) sensor. The aim of the report is to propose a procedure for creating a database on respiration evaluation with the numerical values of the monitored motion of the ribcage, looking at the sixth rib as the most representative one.

In general, respiration is evaluated looking at the respiration frequency, although measures of the ribcage motion are also used. There are three different methods that are currently used to evaluate respiration in terms of respiratory rate, namely using other physiological parameters such as ECG or peripherical blood oxygen saturation, considering movements of the ribcage, and measuring the inhaled or exhaled air flow. Spirometry, as a standard in clinical practice, is used to analyze the respiratory system status, requiring active participation from the subjects under study. An accurate method can be considered a procedure using a combination of accelerometer and gyroscope data to evaluate the respiration with an error of 0.7 respiratory acts [1]. Other methods can be worked out with minor efficiency, like those using optical fibers with an error of 1 respiratory act [1,2] or a piezoelectric–piezoresistive sensor with an error of even 1.8 respiratory acts [1,3]. Different techniques are also tried to monitor respiratory rate by measuring ribcage movements, as, for example, reported in [4–8] from research results. Limitations of these new methodologies are related to high cost, difficulty in practical implementation, and need for an operator during the testing. An interesting new method is based on the SENSIRIB device [9–11] that still requires the presence of an operator during testing. The SENSIRIB device has been used as a basis for the development of the RESPIRholter solution [12], a convenient tool for respiration evaluation in a continuous monitoring of respiratory acts without any operator assistance. In addition, the device's full functionality is accessible by

a patient using RESPIRholter on their own, with a few positive features, such as it being cost effective, lightweight, and comfortable. Indeed, both RESPIRholter and SensiRib are able to collect more data from respiration acts than any other device, for example, angular motion and linear acceleration of the ribcage in terms of the sixth rib. In addition, this numerical measure of the ribcage motion permits an accurate evaluation of the respiration in terms of the biomechanical aspects. In addition, the RESPIRholter device as presented in [12] is a diagnostic device that does not require active participation from the subject under testing and can even be used in non-reactive patients, such as those with dementia or children. Previous experiences with RESPIRholter, as in [13–15], suggest the feasibility of the device for procedures in a campaign for deducing statistical data that can be used for future new diagnostics of respiration acts.

In this paper, the results of implementing RESPIRholter as a device for respiration monitoring during long periods are discussed, deriving from a first pilot experimental campaign involving 10 healthy subjects and 7 patients at the Thoracic Surgery Unit of the University Hospital Policlinico Tor Vergata in Rome. This pilot experimental campaign explores, for the first time, the differences between volunteers and patients who underwent thoracic surgery in terms of numerical values by characterizing respiratory rate and respiratory act motion using angle displacement and acceleration of the ribcage.

## 2. Materials and Methods

A practical evaluation of respiration can be worked out with numerical values by analyzing the motion parameters of the ribcage during respiration in terms of the displacement of ribcage reference points and their motion quality. An experimental evaluation can be performed using the RESPIRholter device in a procedure that can be planned for data collection during long periods and successive data elaboration, like for any other medical holter protocol.

### 2.1. Respiration Characteristics

Respiration is a complex movement that involves the skeletal structure, the muscular complex, and various organs in addition to the pulmonary system, which is its main beneficiary. The resulting movement of the chest is indicative of the functionality and efficiency of the respiratory act, which takes the form of an adequate flow of air in the pulmonary system. Thus, respiration can be monitored and objectively evaluated by analyzing such ribcage motion in terms of both muscle actions and ribcage bone motions. The movement of the ribcage involves both the bone structure and the muscular structure in a synergistic way, determining not only characteristic movements but the consequent expansion and contraction of the lung tissues in precisely allowing for the flow of air during respiration.

The ribcage motion during respiration can be characterized by four basic movements, as summarized in Figure 1a. The so-called pump-handle movement is indicated by the red arrow in Figure 1a, referring to the sternum swinging upward and outward, as produced by the rib system motion. The so-called bucket-handle movement is indicated by the light blue arrow in Figure 1a, and it involves the lower ribs that move laterally and upward like the handle of a bucket. The so-called caliper movement, indicated by the red arrow in Figure 1a, is due to the lowest ribs swinging laterally. A very limited torsion movement is experienced by ribs when twisting around costovertebral joints.

The monitoring and numerical analysis of the movements in Figure 1a can be worked out by looking at the motion characteristics of a representative rib of the ribcage. Main motion data can refer to the movement frequency, angular excursions of the rib, and linear accelerations of a reference rib point as significant kinematic parameters for the quality and quantity of respiration motion. Figure 1b shows a scheme for a measuring system of those motion parameters of a respiratory act by using an IMU (Inertial Measuring Unit) as a detecting sensor in a properly located position on a rib by considering the sensing reference axes. The roll angle is the rotation angle about the X-axis of the sensor reference, whereas

the pitch angle is a rotation around the Y-axis, as shown in Figure 1b. Correspondingly, the acceleration components of the rib reference point are those of the origin of the sensor reference frame. Thus, the accelerometer and gyroscope of the IMU can acquire those data with a proper sampling frequency.

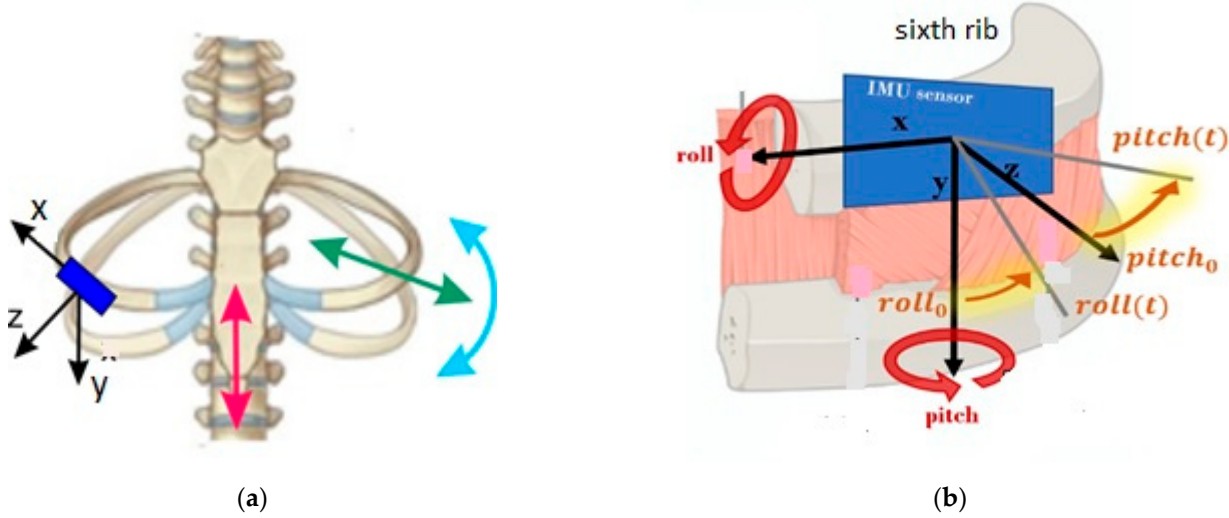

(a)                                                                     (b)

**Figure 1.** Schemes for respiration evaluation: (**a**) main movements in the ribcage; (**b**) monitoring sixth rib. (Colored arrows indicate motion directions).

Such an acquisition of the time data can be evaluated as a characteristic of respiration with values of the respiratory frequency; the angular excursions can be indicative of the respiration efficiency; and the acceleration ranges can be useful in evaluating the quality of the respiratory act, as summarized in Figure 2.

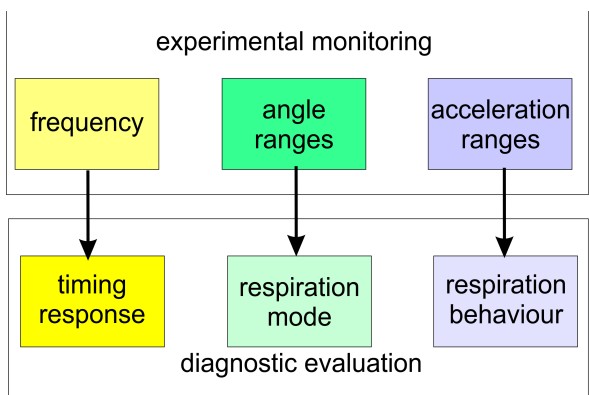

**Figure 2.** A scheme for numerical evaluation of respiration with diagnostic aims.

### 2.2. Respiration Evaluation by Means of RESPIRholter

To acquire rib motion data, a specific device was designed, named RESPIRholter [12,13], whose aim is to acquire specifically acceleration and angle ranges of the VI cost that is considered the most representative of the ribcage motion during respiration. The measured angle ranges reflect the expansion of the ribcage since the more the rib rotates, the more the volume of the ribcage increases with corresponding airflow. The acceleration of a reference point on the rib gives an indication of the smoothness and regularity of respiratory acts while looking at the second order kinematics of the motion to measure respiration behavior not generally but directly appreciated by just vision inspection. Thus, the used IMU as the sensing unit of the RESPIRholter device acquires the acceleration components along the three axes of the reference frame and the roll and pitch angles of the VI rib according to the scheme in Figure 1b within the structure of the device illustrated in Figure 3a. One of

the main features of the RESPIRholter device is summarized in the prototype in Figure 3b showing the small design with a lightweight structure that permits a fairly easy installation and usage.

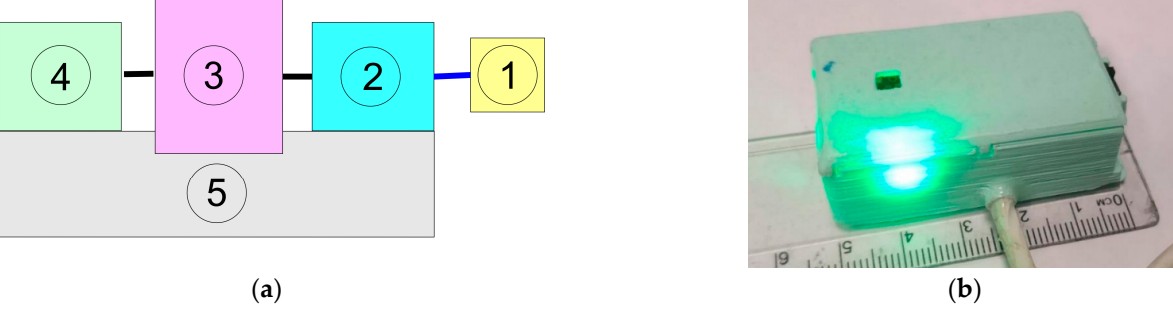

(**a**)  (**b**)

**Figure 3.** The used RESPIRHholter device: (**a**) conceptual design ((1) is motion sensor, (2) is signal acquisition and processing unit, (3) is data storage unit, (4) is connection and data transmission interface, and (5) is a battery); (**b**) a prototype.

The design of RESPIRholter, as reported in [10,12–14], is characterized by its lightweight user-oriented small structure and data elaboration through a predetermined statistical average computation. The design features are shown in Figure 3, and the application on the patient is illustrated in Figure 4.

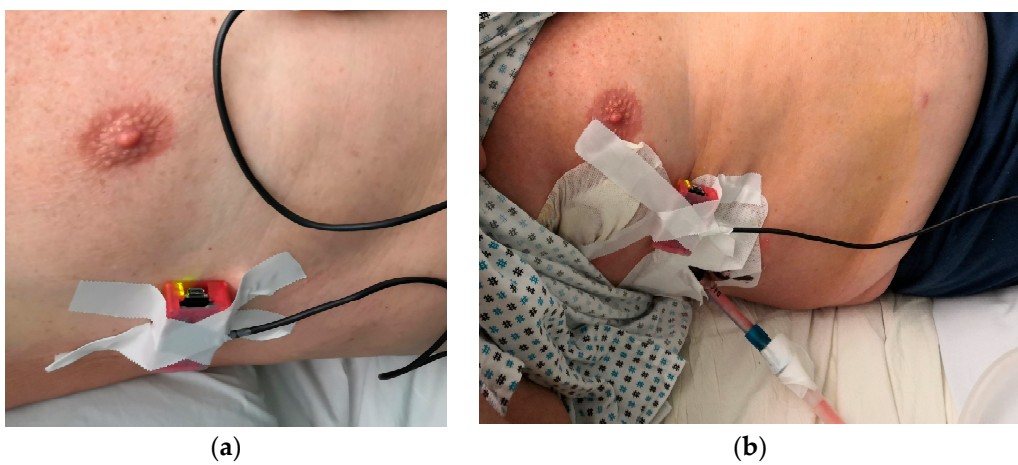

(**a**)  (**b**)

**Figure 4.** An example of installation of REESPIRholter device in a thoracic-operated patient: (**a**) before the operation; (**b**) after the operation.

Referring to Figure 3, the main frame of the RESPIRholter prototype is the box that contains an Arduino NANO 33 IoT board equipped with a 6-axis LSM6DSL IMU sensor. A MicroSD is included as a storage unit, powered by the 5 V power output of the Arduino microcontroller. The device communicates the starting in-operation to the user through blink sequences of a low-power LED with just 1 W of power consumption, including alert of malfunctioning with LED lighting frequency. The power supply is provided by a power bank through a micro-USB connector cable. The RESPIRholter prototype has a size of 55 mm × 30 mm × 16 mm and weighs only 33 g, as shown in Figure 3b.

The small and lightweight design of the RESPIRholter prototype permits an easy installation of the device on the rib VI of a person, either in a healthy person or in a thoracic-operated patient, as shown in Figure 4, with a quite comfortable set up.

*2.3. Procedure for a Campain of Testing*

RESPIRholter was set up to perform a 6 h long operation with acquisition sampling of 1 min each 20 min for each motion parameter. The acquired data are elaborated through a

customized user-friendly RESPIRholter code in MATLAB with a post-processing check that can be adjusted with diagnostic-like comments by a physician. Clicking on the preview option, a physician operator can have a quick look at the acquisitions, and eventually acquisitions can be deleted when they are not considered valid, while on the contrary, when clicking on the execute option, the data acquisition and post-processing elaboration are operated automatically up to a final summarizing report.

The testing procedure is designed together with a specifically designed protocol to standardize the experimental conditions and the activity during testing, as summarized in the step-by-step list in Figure 5.

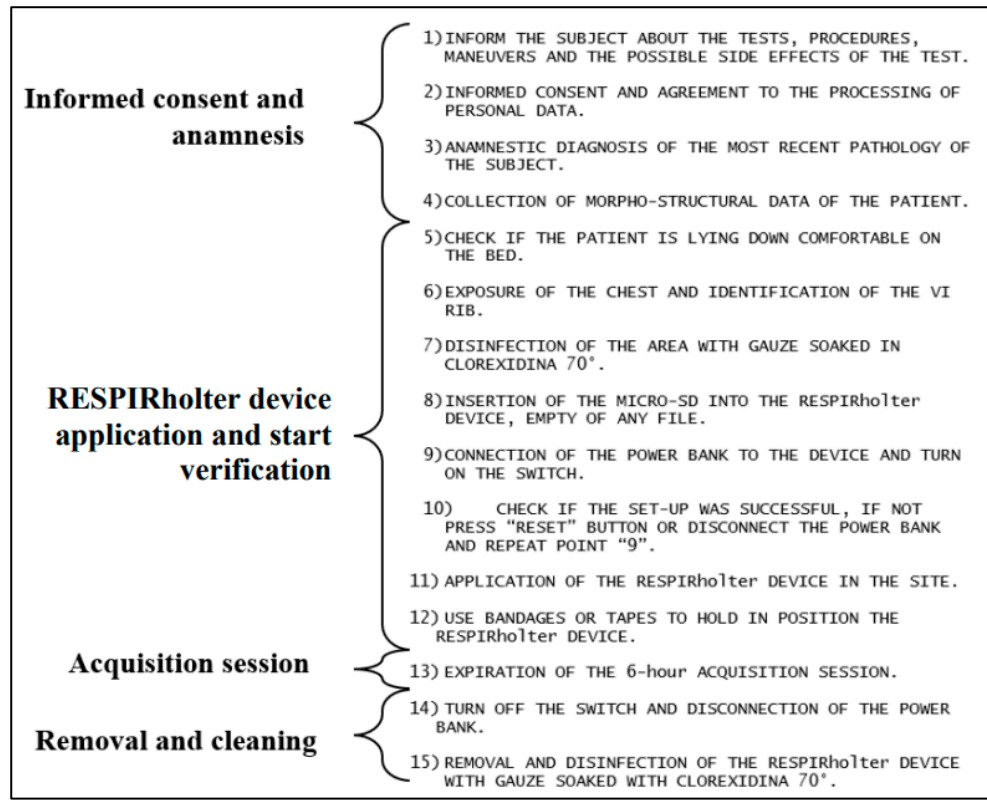

**Figure 5.** A step-by-step procedure of experimental evaluation of respiration.

The first step with points 1 to 4 is to give and collect data for the informed consensus from a patient under testing. The purpose of the testing, the processing of personal data, and how the experiment will be carried out are explained in detail to permit a patient to sign an informed consent form. A brief medical anamnesis interview is then carried out, including the identification of anthropometric data such as height and weight, biographical data, and relevant information about behaviors, diseases, and previous surgeries.

The second step with points 5 to 12 refers to the preparation and application of the RESPIRholter device. The intersection between the anterior axillary line and the sixth rib is identified as the point of application, and the area of the thorax is disinfected with chlorhexidine 2%. Then, the device is prepared and tested once connected to a power bank as to be ready to be fixed with a silk patch, as shown in the example in Figure 4.

The third step with point 13 refers to the acquisition session lasting six hours during the night with the patient lying on the bed.

The last step with points 14 and 15 is run to remove the RESPIRholter device and re-disinfecting the device and the surrounding skin of the patient where the device was installed.

The statistical processing of the acquired data is designed with a data elaboration both to consider the single data segment and the average of all the segments for each of the

motion parameters that are chosen to characterize respiration though the rib motion in terms of angles and acceleration.

In the post-processing data elaboration, the first step is to divide each acquisition into five segments with the same time duration including all the initial and final intervals with all data. Then, the designed RESPIRholter code automatically computes an algebraic average of the five acquisitions for each motion parameter. The acquired data are stored and then visualized in the form of plots. The average of all the acquisitions is obtained next as the algebraic average of the averages yet. In addition, the RESPIRholter code is designed to permit the physician to select a representative plot and a selected plot as per diagnosis purposes in a synthetic representation of the results. An automatic report will contain those main representative plots to facilitate a physician to have a summarized view of the acquired data of the motion parameters during respiration in the 6 h of sampling.

The above procedure is well suited to be used in a campaign of testing with several volunteers to collect results of a comparable value for a statistically significant evaluation of respiration characteristics. Considering the statistical prescription to identify a proper number of samples, as reported in [16], and looking at the possible volunteers of chest-operated patients in a reasonable period of few months, a first campaign was planned to have at least 7 patients as the recurrent number in the report [16], even if not in the same range of age.

## 3. Results

An experimental campaign was planned with a group of 10 chest-operated patients and a group of 10 healthy volunteers to make a comparison of the operation effects and the recovery status after a thorax surgery operation. Unfortunately, only seven patients were finally available with proper characteristics of their chest operation. Each test was conducted following the designed protocol in Figure 5 with satisfactory results referring to the angles and acceleration components of the reference point, including a medical–clinical analysis of the interpretation of the acquired data. In addition, during the testing, several reasons and occurrences, like unexpected malfunctioning due to patients' movements and acquisition stopping for patient walking to the bathroom, the number of valuable acquisitions during the six-hour period were reduced to 10 to have the same number for all the tested persons. Thus, an elaboration of the results was conducted on the best or valid 10 acquisitions of 1 min during the six-hour period.

As an illustrative example, a case study is reported to show the results of the procedure with results valuable in a testing campaign with comparable results among the tested volunteers, either healthy ones or patient ones. The reported case study is also useful to prove that the experimental numerical evaluation of respiration can be of practical significance and even of the expected utility in a clinical diagnosis.

### 3.1. Results of an Illustrative Case of Study

The case study refers to the first test in the campaign and reports results for a 59-year-old man with a BMI of 34.2 who was operated on to undergo pulmonary lobectomy surgery. The data acquisition in his chest-operated condition was worked out during the night between the first and second postoperative day. The RESPIRholter device was installed at the intersection angle between the sixth rib and the anterior axillary line on the left hemithorax, Figure 4, in agreement with the protocol in Figure 5. A pleural drainage was applied during surgery at the fourth intercostal space with no interference with the RESPIRholter device. The patient on the first postoperative day was monitored with good respiration by good peripheral oxygen saturation. On the second postoperative day, peripheral oxygen saturation was down to SpO2 92% (good condition are usually indicated with >94%) so that oxygen therapy was applied in the nasal cannulae with 2 L/min.

The results of the data acquisition are summarized in the plots in Figures 6–8, referring both to direct representative acquisitions and average computation results.

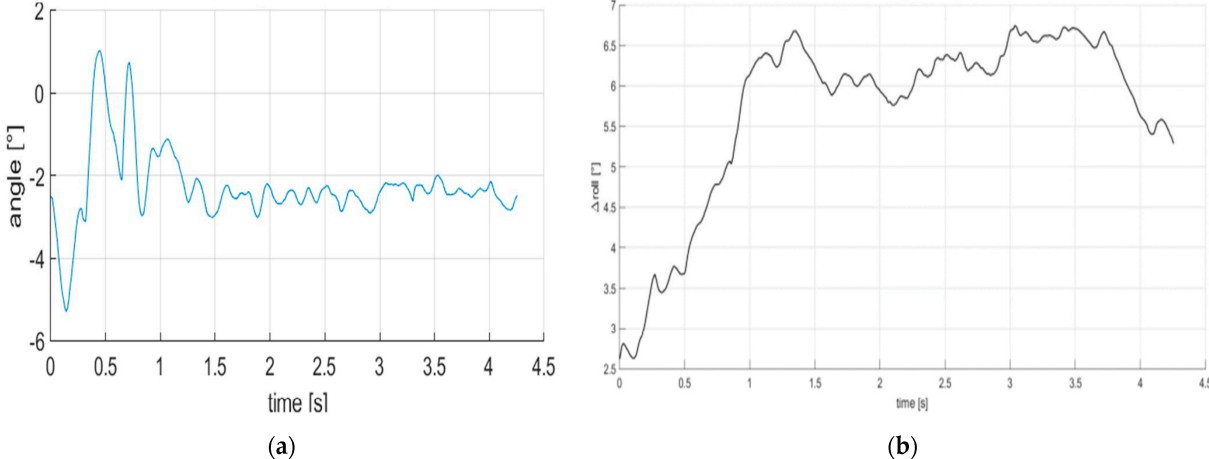

**Figure 6.** Acquired results of rib roll angle during test for the illustrative case study patient: (**a**) for a representative respiration cycle; (**b**) the final average elaboration.

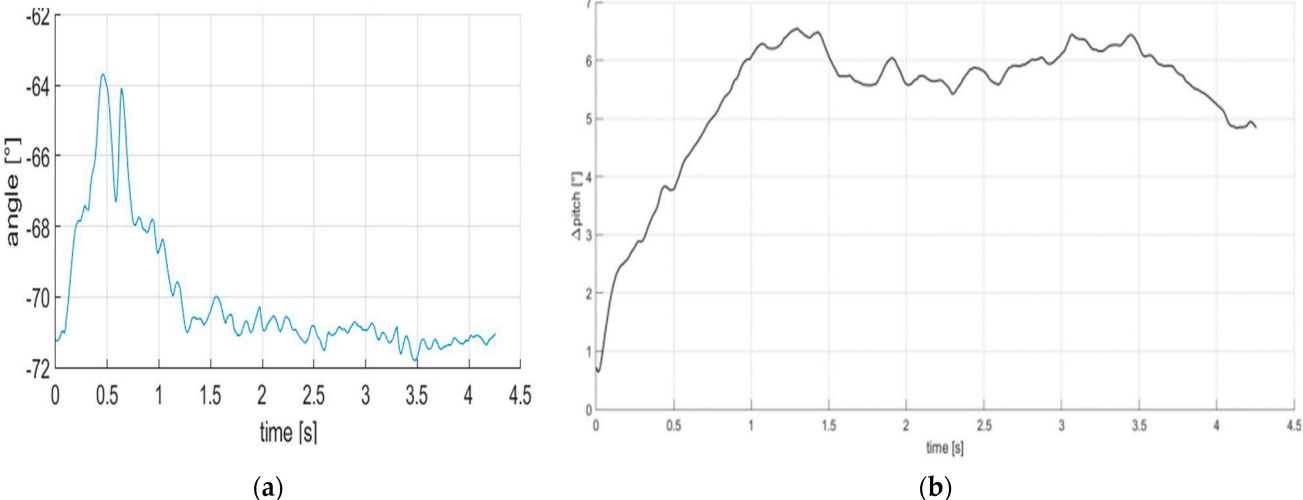

**Figure 7.** Acquired results of rib pitch angle during test for the illustrative case study patient: (**a**) for a representative respiration cycle; (**b**) the final average elaboration.

The results of data acquisition are summarized in the plots in Figures 6–8 referring both to direct representative acquisitions and average computation results. The rib motion is detected by acquiring the angular values and linear acceleration components of the sixth rib at the point of installation of the RESPIRholter box, as in Figure 4. The angular values of roll and pitch describe the overall motion characteristics of ribcage motion in Figure 1a while the acceleration components of the rib point, referring to RESPIRholter box installation, give the second order features of the rib motion that can be used to evaluate its smoothness and regularity.

The elaboration and interpretation of the acquired data have detected respiration of the patient throughout the six hours of acquisition with regular breathing cycles followed by a stage of apnea.

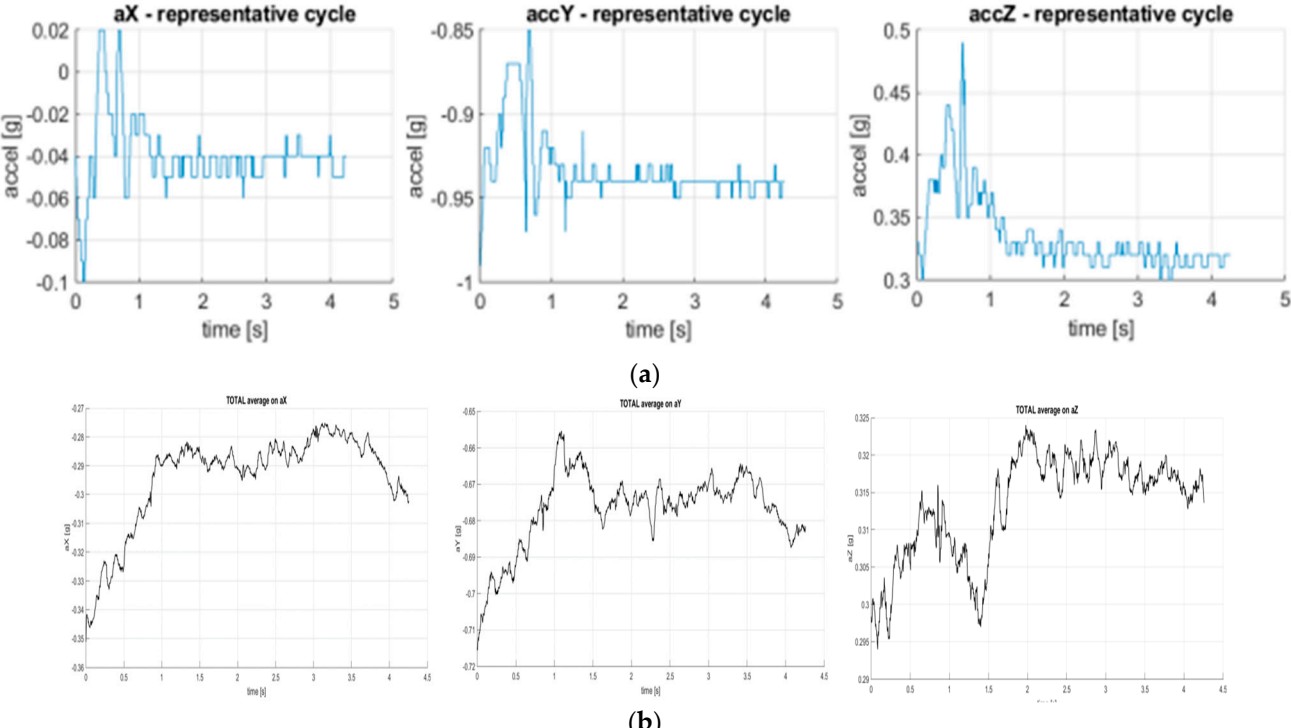

**Figure 8.** Acquired results of rib acceleration components during test for the illustrative case study patient: (**a**) for a representative respiration cycle; (**b**) the final average elaboration.

Figure 6a for the roll angle shows the typical kinematics of respiration with an apnea period that is indicated by the loss of cyclic motion replaced by a quasi-constant motion. The numerical variations in the main motion can be due both to the real functionality of respiration with unsmooth behavior and, mostly, to the compliance of the skin and adipose layer tissue in correspondence to the rib where the IMU is installed. Of clinical significance for the apnea condition is the roll range of the movement which can be noted of about 6 deg in the first part as the regular cyclic part in contrast with the quasi-stationary phase of the apnea. Figure 7b, that is calculated from the average of the average cycles, does not clearly detect such an apnea condition that seems to have few occurrences during the period of the acquisitions, although the effect is evident in a loss of a clear cyclic behavior that is nevertheless characterized by a roll average range of about 6 deg yet.

A similar situation is detected in the plots in Figure 7 representing the acquired pitch motion with the anomaly of the apnea condition that is characterized by a first cyclic part followed by a quite longer quasi-stationary ribcage motion. A gain in the angular variation is a significant parameter, with almost 8 deg in addition to the temporal evolution.

The data acquired in terms of acceleration components, as reported in Figure 8 and Table 1, confirm even more clearly the anomaly of the respiration with apnea. In particular, in Figure 8a, the data acquired with respect to the X-axis show a reduction in accelerations during the stationary apnea phases with respect to the cyclic phase, indicating an excessive stationarity of the respiratory movement up to a minimum range of 0.02 g. In Figure 8b, the few apneas seem not to affect the average values, and cyclic behavior is still observable. Similarly, the Y acceleration component shows a similar reduction in the acceleration variation in the stationary phase up to less than 0.02 g with the average computation with small values due to the more limited motion in general. Even more significant is the result with the acquired acceleration along the Z-axis with a very limited cyclic part and a strongly reduced value for the quasi-stationary apnea period from almost 0.8 g up to less than 0.02 g. The main numerical data are summarized in Table 1 as those deriving from the 10 valid acquisitions during the six-hour period of acquisitions.

**Table 1.** Main characteristic data from 10 segment (seg.) acquisitions for the illustrative case of study in Figures 6–8, reporting the acceleration (a) along the X-, Y-, and Z-axes, with minimum (m) and maximum (M) values, motion angle roll (r), and pitch (p), and respiratory acts per minute (RR).

| Seg. | aXm (g) | aXM (g) | aYm (g) | aYM (g) | aZm (g) | aZM (g) | am (g) | aM (g) | rM (deg) | pM (deg) | RR (act/min) | ERG |
|------|---------|---------|---------|---------|---------|---------|--------|--------|----------|----------|--------------|-----|
| 1 | 0.84 | 0.91 | −0.37 | −0.31 | 0.31 | 0.33 | −0.31 | 0.16 | 0.89 | 1.05 | 12 | 8 |
| 2 | 0.84 | 0.89 | −0.35 | −0.32 | 0.28 | 0.34 | −0.47 | 0.05 | 1 | 1.24 | 12 | 8 |
| 3 | 0.81 | 0.87 | −0.5 | −0.44 | 0.23 | 0.25 | −0.20 | 0.06 | 1.03 | 1.23 | 12 | 8 |
| 4 | −0.75 | −0.69 | 0.25 | 0.31 | 0.59 | 0.66 | −0.13 | 0.04 | 1.47 | 1.54 | 12 | 8 |
| 5 | −0.75 | −0.69 | 0.28 | 0.31 | 0.6 | 0.66 | 0.00 | 0.04 | 1.66 | 1.75 | 12 | 8 |
| 6 | 0.77 | 0.81 | −0.34 | −0.28 | 0.5 | 0.56 | −0.21 | 0.23 | 0.49 | 1.31 | 12 | 8 |
| 7 | 0.85 | 0.85 | −0.49 | −0.47 | 0.2 | 0.22 | 0.01 | −0.04 | 0.54 | 0.84 | 12 | 8 |
| 8 | −0.68 | −0.66 | 0.19 | 0.25 | 0.66 | 0.72 | −0.33 | 0.08 | 0.95 | 1.06 | 12 | 8 |
| 9 | −0.72 | −0.66 | 0.22 | 0.25 | 0.66 | 0.68 | 0.01 | −0.20 | 0.95 | 1.39 | 12 | 8 |
| 10 | −0.52 | −0.5 | 0.09 | 0.16 | 0.81 | 0.85 | −0.33 | −0.01 | 0.7 | 1.35 | 12 | 8 |

### 3.2. Statistic Elaboration of Testing Results

A statistical elaboration of the data results during this first campaign from tests with 7 chest-operated patients and 10 healthy volunteers is reported in Figures 9–11.

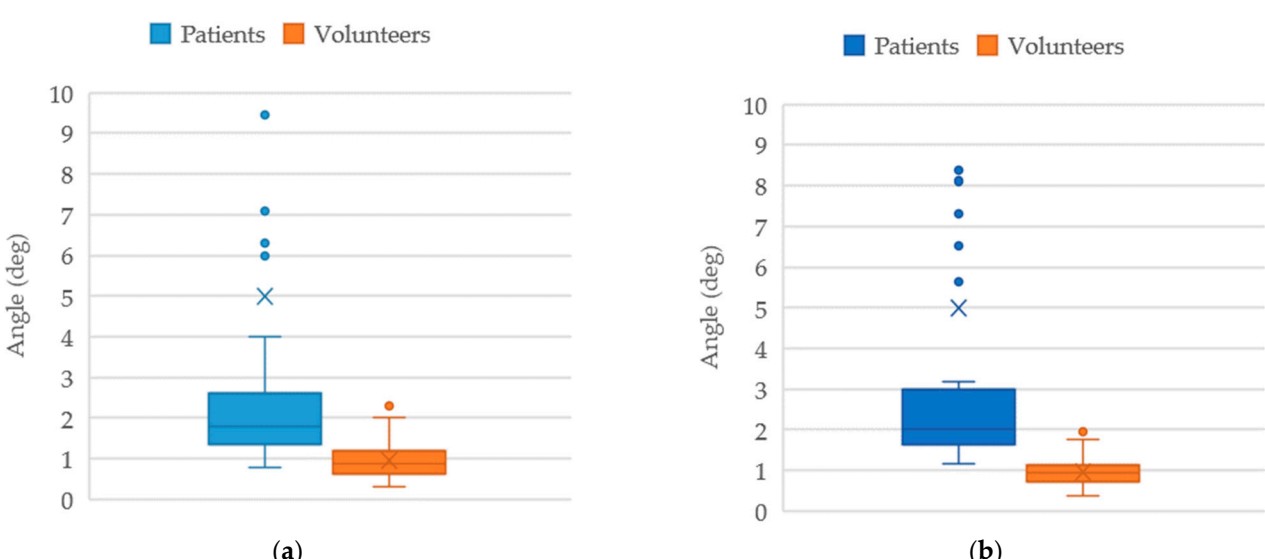

**Figure 9.** Box plot of data from healthy volunteers and patient acquisitions: (**a**) roll angle; (**b**) pitch angle.

Descriptive statistics have been performed for healthy volunteers or patients after surgery. With these first campaign results, the sample size does not consent the execution of a parametric statistical elaboration. In Figure 9a, the average roll motion shows a difference from 0.97 deg in healthy people to 4.99 deg in patients after surgery, with a much larger standard deviation. In Figure 9b, the average pitch motion gives a difference from 0.96 deg in healthy people to 5.00 deg in patients after surgery, again with a much larger variability in behavior. This indicates that the operated patient shows a respiration motion with much maximal-type breathing and larger rib motion than in healthy people, as indicated by the value of increased average and range.

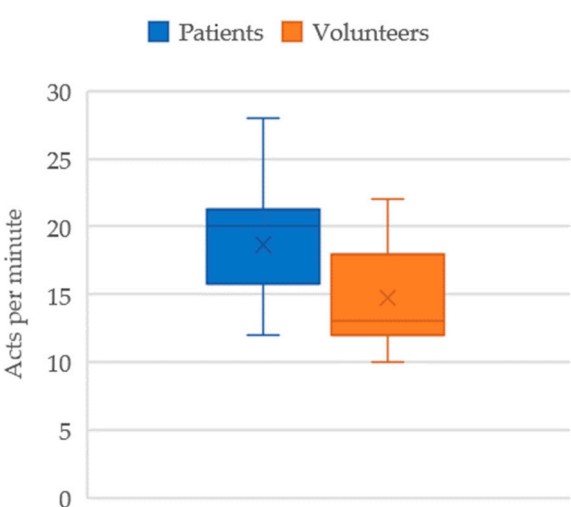

**Figure 10.** Box plot of respiratory rate from healthy volunteers and patient acquisitions.

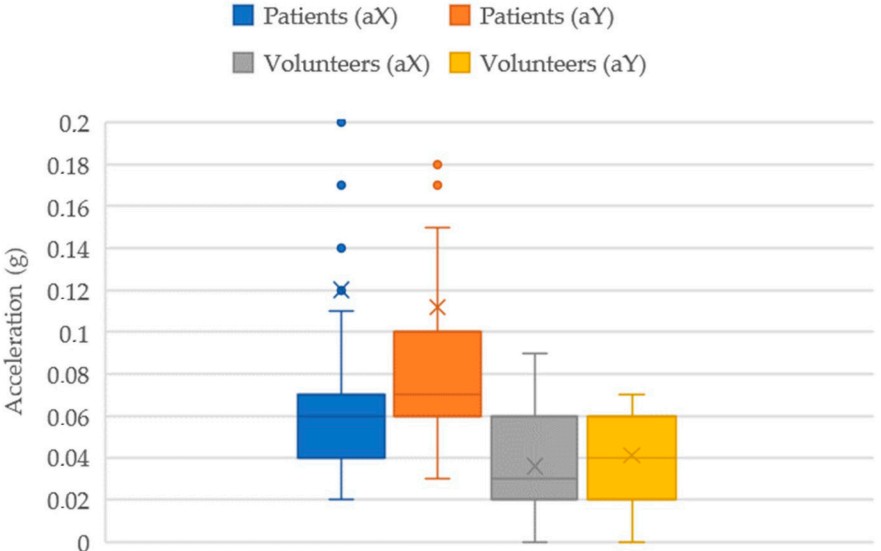

**Figure 11.** Box plot of X and Y acceleration components from healthy volunteers and patient acquisitions.

This behavior is also reflected in the respiratory rate statistics in Figure 10, which shows a difference from 13 acts/min in healthy volunteers (that is in agreement with the literature as in [1]), to 20.00 acts/min in patients after surgery. The respiratory rate is, therefore, increased in patients after surgery who appear to be tachypneic, as indicated by the larger value of the average in addition to the higher standard deviation.

In Figure 11, the statics elaboration results are reported, referring to the variations in the X and Y acceleration components neglecting the Z component that does not show significant variation. The X and Y acceleration components in standard deviation remain quite similar in healthy volunteers, whereas the average is detected to be almost equal. On the contrary, in patent volunteers the Y acceleration components there is a much larger deviation with an increased average with respect to X components.

The boxplots in Figure 11 indicate that healthy volunteers show a smooth ribcage motion both on the X-axis (laterally) and Y-axis (downward), whereas patients after surgery show a staggered downward (Y) motion when compared to the lateral (X) component. The results indicated that, in general, the patients after surgery performed respiration with a ribcage motion that is characterized by deeper and faster breathing, similar to a maximal regular type with ragged breathing.

In summary, the first campaign results provide a first database of respiration statistic characteristics of patients after surgery with values that have been used for the comparison of the data of healthy volunteers to show the soundness and feasibility of the proposed database and procedure yet. In particular, clinical implications can be recognized for the possibility of creating a database of reference values of the rib motion characteristics and to use this database as diagnostic reference for detecting regular respiration, but anomalies in the recovery of thorax-operated patients, likewise in other clinical procedures. The database is limited with the current results that nevertheless give indications of a promising accumulation of those data when proper consistent experimental campaigns are carried out in terms of a different category of subjects, such as age, gender, and BMI characteristics. Thus, the presented results give a clear indication for the proposed testing procedure to be used in future work with the use of the RESPIRholter device and statistical data post-processing with an adequate number of subjects both in good health and in post-operated conditions.

## 4. Conclusions

This paper presents the results of a first campaign of testing by comparing the data of an experimental respiration evaluation from healthy people and thorax-surgery-operated patients. The reported results show clear numerical values that can be used for diagnostic purposes. Indeed, the aim of the work is to propose a procedure that can be useful to create a suitable database with the selected numerical parameters of ribcage motion during respiration using the biomechanical movement of the sixth rib. A first database can be considered with the results of this first testing campaign, giving a clear indication of the aspects for future work in the proper selection of tested subjects with proper common characteristics and conditions, and in plans for more repeated tests to be carried out during the pre- and postoperative period.

## 5. Patents

Marco Ceccarelli, Portable device for measuring the movement of human ribs, Patent no. IT. 102021000005726, 21 March 2023.

Marco Ceccarelli, Vincenzo Ambrogi, Lucrezia Puglisi, Matteo Aquilini, Ventilatory holter device, Patent no. IT. 102021000008585, 18 April 2023.

**Author Contributions:** Conceptualization, M.C. and V.A.; methodology, M.C., M.D., V.A. and M.R.; software, M.D. and M.R.; validation, M.D. and M.R.; formal analysis, M.C., M.D. and V.A.; investigation, M.C., M.D., V.A. and M.R.; resources, M.C. and V.A.; data curation, M.D. and M.R.; writing—original draft preparation, M.C. and M.D.; writing—review and editing, M.C., M.D., V.A. and M.R.; visualization, M.C. and M.D.; supervision, M.C. and V.A.; project administration, M.C.; funding acquisition, M.C. All authors have read and agreed to the published version of the manuscript.

**Funding:** This research received no external funding.

**Institutional Review Board Statement:** The study was conducted in accordance with the Declaration of Helsinki and approved by the Institutional Ethics Committee of Policlinico di Tor Vergata, Rome, with protocol code RS. 197.22 on 15 November 2022.

**Informed Consent Statement:** Informed consent was obtained from all subjects involved in the study. Written informed consent has been obtained from the patients to publish this paper.

**Data Availability Statement:** Not applicable.

**Conflicts of Interest:** The authors declare no conflict of interest.

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
