# Peer review of "An Experimental Evaluation of Respiration by Monitoring Ribcage Motion"

_applsci, doi:10.3390/app13158938_

Round 1
Reviewer 1 Report
In this manuscript, the authors address the problem of a statistically significant numerical evaluation of the biomechanics of respiration using the motion of the ribcage through the identification of the kinematics of the sixth rib. The authors present results of a first campaign of testing by comparing data of an experimental respiration evaluation from healthy people and thorax operated-surgery operated patients. The results show clear numerical values that can be used for diagnostic purposes. In general, the authors present lots of results, and my assessment of this manuscript is positive. My biggest concern is that the resolution of the figures is low. So, I think the authors should address this issue prior to publication in applied sciences.
Author Response
pls find attacedh the reply file

Reviewer 2 Report
The article, "An experimental evaluation of respiration" by Marco Ceccarelli et al., presents promising results in the evaluation of respiration biomechanics using rib cage motion. However, several areas need improvement in terms of clarity, methodology description, and discussion of the results.
1. The title of the paper should be more specific about the experimental evaluation being performed. Consider changing the title to reflect the emphasis on respiratory biomechanics and the use of rib cage motion.
2. The abstract should provide a more concise summary of the study. It should clearly state the research objective, methodology, and key findings.
3. In methodology, include detailed information about the RESPIRholter device prototype used for monitoring and data collection.
4. In results, provide more specific details about the statistical processing of the data and the nature of the acquired motion characteristics. Include more details about the statistical tests that were used and the significance of the reported values. Provide clear descriptions of the plots and movement graphs presented.
5. Discuss clinical implications, limitations, and future research in the discussion.
6. Ensure clarity, language, and organization throughout the paper.
By incorporating the suggested improvements, the article can strengthen its contribution to the field and provide readers with more comprehensive insights.
Author Response
pls find attacedh the reply file

Reviewer 3 Report
The paper is very interesting, with substantiation and valuable experimental results, and it is recommended to continue the research for the practical implementation of the obtained results.
Author Response
pls find attacedh the reply file
